# Knowledge about Asymptomatic Malaria and Acceptability of Using *Artemisia afra* Tea among Health Care Workers (HCWs) in Yaoundé, Cameroon: A Cross-Sectional Survey

**DOI:** 10.3390/ijerph20136309

**Published:** 2023-07-06

**Authors:** Abenwie Suh Nchang, Lahngong Methodius Shinyuy, Sandra Fankem Noukimi, Sylvia Njong, Sylvie Bambara, Edgar M. Kalimba, Joseph Kamga, Stephen Mbigha Ghogomu, Michel Frederich, Jean Lesort Louck Talom, Jacob Souopgui, Annie Robert

**Affiliations:** 1Department of Epidemiology and Biostatistics (EPID), Institute de Recherche Expérimentale et Clinique (IREC), Public Health School, Université Catholique de Louvain (UCLouvain), 1200 Brussels, Belgium; sylvie.njong@student.uclouvain.be (S.N.); sylvie.bambara@amphealth.org (S.B.); annie.robert@uclouvain.be (A.R.); 2Laboratory of Pharmacognosy, Department of Pharmacy, Center of Interdisciplinary Research on Medicine (CIRM), University of Liege, 4000 Liège, Belgium; ms.lahngong@doct.uliege.be (L.M.S.); m.frederich@uliege.be (M.F.); 3Embriology and Biotechnology Laboratory, IBMM-ULB, 6041 Gosselies, Belgium; sandra.fankem.noukimi@ulb.be (S.F.N.); jacob.souopgui@ulb.be (J.S.); 4King Faisal Hospital, Kigali 2534, Rwanda; edgar.mk@kfhkigali.com; 5Biotechnology Unit, University of Buea, Buea P.O. Box 63, Cameroon; josephkamga700@yahoo.com (J.K.); stephen.ghogomu@ubuea.cm (S.M.G.); 6Aumônerie—Hôpital du Jura bernois SA, 2740 Moutier, Switzerland; jean.louck@hjbe.ch

**Keywords:** health care workers, asymptomatic *Plasmodium* carriers, knowledge, *Artemisia afra* tea, treatment

## Abstract

Malaria is the most widespread endemic disease in Cameroon, and asymptomatic *Plasmodium* (gametocyte) carriers (APCs) constitute more than 95% of infectious human reservoirs in malaria endemic settings. This study assesses the knowledge of asymptomatic malaria (ASM) among health care workers (HCWs) in health facilities (HFs) in the Centre Region of Cameroon and the acceptability of using *Artemisia afra* tea to treat APCs. A cross-sectional descriptive survey was conducted among 100 HCWs, in four randomly selected HFs in the Centre Region, in the period of 1–20 April 2022, using semi-structured self-administered questionnaires. Logistic regression analyses were performed to determine factors associated with knowledge. More than seven in eight (88%) respondents were aware of the existence of ASM, 83% defined ASM correctly, 75% knew how it was diagnosed, 70% prescribe ACTs for APCs, and 51.1% were informed about ASM transmission. The professional category of HCWs was significantly associated with their knowledge of the existence and transmission of ASM, and longevity of service was associated with knowledge of transmission (*p* < 0.05). Two-thirds (67%) of respondents knew about *Artemisia afra* tea, 53.7% believed that it was effective in treating ASM, and 79% were willing to prescribe it if authorized. There was a fair level of knowledge of ASM among HCWs in the study settings.

## 1. Introduction

Malaria is a preventable, curable, and life-threatening disease caused by the *Plasmodium* parasite Laveran, 1980, and it is transmitted by female *Anopheles* mosquitoes Meigan, 1818 [1]. There are five Plasmodium parasite species that cause malaria in humans including *P. falciparum*, *P. vivax*, *P. malariae*, *P. ovale*, and *P. knowlesi*. Of these species, *P. falciparum* and *P. vivax* pose the greatest threat, and *P. falciparum* is the deadliest and the most prevalent on the African continent, while *P. vivax* is dominant in most countries outside of sub-Saharan Africa. In Cameroon, four *Plasmodium* species are presently known to cause malaria (except *P. Knowlesi*), among which *P. falciparum* is responsible for 82–100% of all malaria cases. While the four other Plasmodium parasites species are transmitted between humans by mosquito vectors belonging to the genus *Anopheles*, it has become evident that *P. knowlesi*, which typically infects forest macaque monkeys, can be transmitted by anophelines to cause malaria in humans in South East Asia (presently considered a zoonotic disease) [2]. According to the WHO world malaria report 2022, there were 247 million malaria cases and 619,000 deaths worldwide [3] in 2021, compared to 245 million cases and 625,000 deaths in 2020 [1]. Of these, the WHO African Region carries an excessively high share of the global burden, with 95% of cases and 96% of malaria deaths [1]. Cameroon was among 11 countries that had the highest burden of malaria in WHO African region in 2020, with 2.9% of all regional malaria cases, and 2.4% of malaria deaths [3]. Likewise, Cameroon was among three countries that shouldered over 80% of malaria cases and deaths in the Central African sub-region in 2020, representing 12.6% of subregional malaria cases and 11% of deaths. The WHO strategic framework for malaria elimination recommends effective vector control; prophylactic treatment (vaccination and chemoprophylaxis) for vulnerable populations; and accurate diagnosis (passive and active case detection) and treatment of acute symptomatic persons and asymptomatic gametocyte carriers [4]. 

Repeated calls from the WHO advocating for mobilization and expanded actions towards malaria elimination in the continent have resulted in an extraordinary and optimistic scale-up of WHO-recommended malaria control interventions in Cameroon, as in most African countries, over the last few decades. The current measures in place include vector control (insecticide-treated nets (ITNs) and indoor residual spraying (IRS)), early diagnosis and prompt treatment of symptomatic cases, as well as chemoprophylaxis for high-risk population groups such as children under 5, pregnant women and naïve visitors [5]. Despite the recent progress on reducing malaria morbidity and mortality in response to this, empirical and theoretical evidence shows that current control interventions alone will not be sufficient to eliminate malaria from many endemic regions in sub-Saharan Africa, thereby suggesting their supplementation with additional strategies, for effective elimination [6]. The first reason is the change in climate, which has increased the prevalence of malaria and other vector-borne diseases in recent decades, with the risk of a further increase if control measures are not intensified [7]. A second reason is that the effectiveness of the current strategies has been threatened by the emergence of resistance to both insecticide and antimalarial drugs such as ACTs, as reported in several African countries [8,9,10]. In addition, these strategies are restricted to indoor controls, and populations remain exposed to *Plasmodium* transmission during their outdoor activities. Furthermore, these strategies focus on the mosquito stage and the two asexual human stages of the *Plasmodium* parasite, but they fail to target the gametocyte stage in humans, which facilitates the parasite transmission to the mosquito vector for the disease perpetuation; meanwhile, asymptomatic *Plasmodium* reservoirs (gametocyte carriers) constitutes more than 95% of infectious human reservoirs in endemic settings, and only a small proportion of infectious reservoirs are symptomatic [11].

Gametocytes are insensitive to most conventional antimalarial drugs, and often persist in the blood stream after successful antimalarial therapy, thereby ensuring continued malaria transmission over several months after the clearance of asexual parasites [12,13]. This results in the widespread occurrence of asymptomatic *Plasmodium* carriers, who often do not seek antimalarial treatment, due to the absence of clinical illness. Hence, there is a need for health care workers to consciously engage in accurate diagnosis and effective treatment of asymptomatic carriers, to prevent continuous transmission. 

Primaquine and the fast-acting but short-lived artemisinin-based drugs are the only antimalarials that have exhibited gametocytocidal activities [12,14], but primaquine is toxic in persons with glucose-6-phosphate dehydrogenase deficiency (G6PD), limiting its general use in most countries [15,16]. A novel antimalarial compound called Cipargamin (KAE609) compound, which is said to exert gametocidal activity with transmission-blocking potential, is still undergoing phase 2 clinical trials to assess its safety and efficacy [15].

On the other hand, studies have demonstrated that *Artemisia annua* Tu, 1972 and *Artemisia afra* Huxley et al., 1992 in the form of infusions or powdered leaves reduce parasitemia and might kill the transmissible gametocyte forms, with characteristic chemoprotective potentials [17,18,19,20,21,22,23]. 

*Artemisia annua*, also referred to as sweet wormwood, sweet annie, or annual wormwood, is an aromatic herb that originated in China, but it is being widely distributed throughout the world due to its ability to treat a wide range of diseases, traditionally speaking [24]. It has been used in China since ancient times as a herbal product taken in the form of tea infusions or decoction for the alleviation of intermittent fevers associated with malaria, amongst other diseases [25]. It is now domesticated and used in other parts of Asia, Europe, Australia, and America, to fight other ailments [25,26,27,28]. For decades, the phytochemical investigations of *A. annua* has been at the forefront of research due to the isolation of artemisinin, a potent anti-malarial compound [29]. Artemisinin has been found to have a remarkable efficacy against the erythrocytic stages of the malaria parasite, and several in vitro and in vivo studies as well as clinical trials on the crude plant material and/or tea infusions have been reported, showing good anti-plasmodial activities [30,31]. To this effect, the WHO recommends “Artemisinin Combination Therapy (ACTs)” as the first line of treatment for uncomplicated malaria due to *Plasmodium falciparum* [3,32]. However, the WHO has cautioned against the use of non-pharmaceutical sources of artemisinin because of the risk of delivering sub-therapeutic doses of artemisinin that could potentiate anti-malarial drug resistance [32]. 

*Artemisia afra*, also referred to as African wormwood, is a herb indigenous to Africa and a member of the same genus as *A. Annua*. It is native to South Africa and has a long history of use as a traditional remedy against coughs, colds, fever, and malaria [33]. It is cultivated across the African continent for the management of malaria and other diseases [34].

While *A. annua* cultivars have high artemisinin content, cultivars of *A. afra* are devoid of artemisinin, while other cultivars have negligible contents of artemisinin [17]. Despite it being devoid of artemisinin or having a negligible artemisinin content, *A. afra* has also demonstrated good anti-plasmodial properties both when it is used in infusions or crude extracts in vitro and in vivo [17,18,19,21,22,30,35,36,37]. In vitro studies have also confirmed antiparasitic activity of *A. afra* tea infusions against the different stages of the malaria parasite including the gametocyte stage, with no cytotoxicity [17,18,19,21,22].

The reports that *Artemisia afra* tea infusions effectively treats malaria, with gametocidal effects and no reported toxicity [17,18,19,21,22], suggest its probable usefulness as an alternative treatment to ACT in the elimination of *Plasmodium* reservoirs in endemic settings. However, the opinions and willingness of HCWS to prescribe or use *A. afra* tea to treat asymptomatic gametocyte carriers has not been ascertained. In respect of the WHOs recommendation to integrate traditional medicine in national health care systems, the Cameroon government wants to promote the use of traditional medicine and is resolute on encouraging the treatment of patients with alternative medicine from traditional sources [38]. The objective of this study was therefore to assess the knowledge of asymptomatic malaria among HCWs in some rural and urban health facilities in and around Yaoundé, in the Centre Region of Cameroon, and to explore their opinions and the acceptability of using *A. afra* tea infusions to treat asymptomatic *Plasmodium* carriers. 

## 2. Materials and Methods

Study design: A cross-sectional descriptive survey was conducted among 100 health care workers, to assess their knowledge of asymptomatic malaria and their acceptability of using *A. afra* tea to treat asymptomatic *Plasmodium* carriers.

Study area and sites: The study was conducted in Yaoundé and its environs, in the Centre Region of Cameroon. The Center Region is one of the 10 administrative regions of Cameroon which lies in the southern plateau. It is one of the most densely populated regions of Cameroon, with a population of over 3 million inhabitants mainly living in the city of Yaoundé (population 1.1 million) or along the roads and in the major towns [39]. The Center Region is in-turn divided into 10 administrative divisions (Figure 1), with a total of 30 health districts (7 urban and 23 rural). It is serviced by many hospitals and clinics, particularly in Yaoundé and in the larger towns. Besides being the headquarter of the Centre Region, Yaoundé is also the capital city of Cameroon and the second largest city of Cameroon [40]. The city is located within the Congo-Guinean phytogeographic zone, characterized by a typical equatorial climate with two rainy seasons extending from March to June and from September to November. It is situated 800 m above sea level and surrounded by many hills [41]. Malaria transmission in Yaoundé is considered to be holo-endemic and seasonal [42,43]. Meanwhile, the median annual prevalence of *Plasmodium falciparum* is estimated to vary between 34 and 50% in the general population from the city center to the periphery [44]. The most affected people are children between 0 and 15 years old, comprising 75% of asexual parasite carriers, 85% of carriers of high parasitemia and 83% of gametocyte carriers [45]. 

The survey was conducted in four district hospitals in the Mfoundi and Lekie divisions of Centre Region (Yaoundé) of Cameroon, between January and April 2022. These included two (2) urban hospitals in the Mfoundi (Biyem-Assi and Efoulan district hospitals) and two (2) rural hospitals in Lekie (Obala and Okola districts hospitals) (Figure 1).

Study population: The study population consisted of HCWs aged from 18 to 65 years; those who had ever been involved in patient consultation and prescription of treatment and were currently working; and those who were working in the study hospitals during the study period. They were mainly physicians (general practitioners and specialized doctors) and nurses (degree nurses, state registered nurses, and nursing assistants). HCWs who had never been involved with patient consultation or treatment prescription, and those who were interns, were excluded.

Sampling method: Clustering. Four health districts were randomly selected from the thirty health districts in the Center Region of Cameroon (2 among the 23 rural health districts and 2 among the 7 urban health districts) by lottery method, in which numbers were assigned to the health districts involved, and random numbers were drawn from the pool. The district hospital of selected health district was systematically enrolled. A total of 100 HCWs were recruited in the four study sites. All HCWS who met the inclusion criteria and consented to participate were included in the sample. Eligible participants were consecutively recruited and interviewed until the pre-established sample size of 25 HCWs per facility was reached. 

The study instrument: The questionnaire was developed based on the study objectives, while the questionnaire design and the choice of questions was informed by the available literature from previous studies, reviews, and reports on the knowledge of HCWs about malaria management. The questions were designed to have a clear and consistent format, with clear and concise instructions, made simple and easily understandable. The questionnaire had three sections addressing socio-demographic information, knowledge of HCWs on asymptomatic malaria and their opinion and acceptability of using *Artemisia afra* tea for patients’ treatment. The questions were both open-ended and close-ended. The open-ended questions were used to capture HCWs’ perceptions and assess their knowledge on the definition of asymptomatic malaria, how it is diagnosed and treated, the necessity for its treatment, and its transmissibility. The questions were also used to obtain information on the uses of *Artemisia afra* tea in this context, the perception and opinions of HCWs on its use to treat *Plasmodium* reservoirs, the acceptability of prescribing these teas if recommended by authorities, and the possible challenges of using these teas to treat asymptomatic *Plasmodium* carriers. The instrument allowed participants to give multiple responses to the open-ended questions. 

Data collection procedure: Data were collected using semi-structured self-administered questionnaires we developed for the purpose of the study. The questionnaires were completed in the presence of research assistants who were trained for the questionnaire administration to ensure that the questionnaires were well completed and returned on time.

Data quality control: Intensive training was provided to research assistants (data collectors) on data collection techniques. Detailed orientation was given to the data collectors about the study before the start of data collection procedure. The questionnaires was pre-tested on 10 health care workers in a hospital aside of the study, to ensure that respondents clearly understood the objectives of the questions and provided appropriate responses to the questions, and to validate consistency of the questions and collection tool. Questions that were identified as not clear were rephrased to improve comprehension.

Study variables: The outcome variables were: knowledge of asymptomatic malaria among HCWs (existence, definition, diagnosis, management, and transmission); opinion and acceptability of *Artemisia afra* tea among HCWs and predicted challenges of using the tea to treat asymptomatic *Plasmodium* carriers; and factors associated with knowledge of asymptomatic malaria among HCWs, such as age of HCW, gender, professional category, longevity of service, and site of practice.

Ethical considerations: The study was conducted according to the guidelines of the Declaration of Helsinki and approved by the Faculty of Health Sciences-Institutional Review Board (FHS-IRB), University of Buea, Cameroon (2021/1591-01/UB/SG/IRB/FHS). A conditional ethical approval was also obtained from the ethics committee of the UCLouvain medical faculty, Brussels, Belgium (Comite d’Ethique Hospitalo-Facultaire/8 November 2021). Administrative authorization was obtained from the relevant authorities of each study site. Each participant in the study was provided with necessary information about the study including the respect of confidentiality and their right to voluntarily participate. Informed consent was obtained from all subjects involved in the study. They were allowed to discontinue the interview when they needed. 

Data management and statistical analysis: Results are presented using numbers with percentages for categorical data. Qualitative content analysis was used to analyze the open-ended questions, and frequencies of occurrences were quantified to complement the quantitative data. Participants who gave correct a response for open-ended questions on knowledge assessment were considered to be knowledgeable, while those who either gave the wrong response or said “I don’t know” were considered not knowledgeable for that item, and those who did not answer were considered as missing data. Simple and multiple logistic regression analyses were performed to assess associations of respondents’ knowledge of existence, definition, diagnosis, and transmission of asymptomatic malaria with explanatory variables such as age, gender professional category, longevity of service, and practicing site status. Explanatory variables with a *p*-value lower than 0.25 in the univariate analysis were introduced in a multivariate logistic regression model using a backward stepwise elimination procedure with likelihood ratio chi-square test. Odds ratios and their 95% confidence interval are reported. Analyses were performed with Stata version 17.0 software in HP elitebook Folio970 m laptop, (by Hewlett -packard company Califonia and sourced from Oxfarm-Brussels, Belgium) and a *p*-value < 0.05 was considered as statistically significant. 

## 3. Results

### 3.1. Characteristics of Survey Participants

Table 1: Among the 100 health care workers (HCWs) who participated in this survey, 75% of them were at least 30 years old, and 80% of them were women. A majority (72%) of participants were nurses of various grades, 28% were medical doctors (general practitioner, family doctors or specialists in medicine), and 49% of them had practiced for five years or fewer.

### 3.2. Knowledge of Asymptomatic Malaria among Health Care Workers

Table 2: Among the survey participants, 88 (88%) were aware of the existence of asymptomatic malaria, and 12% had never heard about asymptomatic malaria before. Of these, 73/88 (83.0%) defined asymptomatic malaria correctly as the presence of malaria parasites in the blood without symptoms, and 66/88 (75%) of them knew that asymptomatic malaria was diagnosed through microscopic blood examination. Most of respondents (75/88; 85.2%) had encountered patients with asymptomatic malaria, and 70/75 (93%) put the patients on treatment including ACT (49/70; 70%), artesunate (47.1%), quinine sulphate (17.1), and others (44.2%): fansidar, camoquine, halfantrim, suquina, artecom, malarone, doxycycline, vibramycine, oraceal, mefloquine, and tefaroquine.

Most HCWs (71/88; 80.7%) were aware that it was necessary to treat asymptomatic *Plasmodium* carriers but did not know the reason for treating, as they said the reason for treating carriers was to avoid severe complications (36/88; 40.9%), to prevent development of symptoms (26.1%), and due to the patient living in a malaria endemic zone (11.4%); only 2 (2.3%) respondents gave the correct reason that was to curb transmission. Meanwhile, 9/88 (10.2%) respondents thought that it was not necessary to treat asymptomatic carriers, and 8 (9.1%) did not know if asymptomatic carriers needed to be treated or not. In a similar manner, 86.4 % of respondents agreed that it was risky to leave asymptomatic carriers untreated, but they did not know the correct risk involved, as the majority of them said if left untreated, carriers were at risk of developing complications (52/88; 59.1%) or becoming symptomatic (18.2%); and only 3 (3.4%) respondents mentioned risk of transmission. Meanwhile, 4.5% of the respondents thought that there was no risk if asymptomatic carriers were not treated, and 9.1% did not know if there was or was not a risk to treat (Table 2).

More than half (45/88; 51.1%) of respondents were aware that healthy people could become infected by *Plasmodium* parasites transmitted from asymptomatic carriers, while 44.4% mistakenly thought that this was not possible because asymptomatic malaria is not transmissible (35.2%) and because the parasite quantity (load) in asymptomatic carriers is insignificant (8.0%). Still on transmission, 29/55 (89.1%) respondents were aware that a person infected with *Plasmodium* parasites from an asymptomatic carrier could develop malaria symptoms because the parasite was present (52.7%) and because immune systems are different (36.4%), while 10.9% of participants did not know if this was possible or not (Table 2).

### 3.3. Knowledge of Asymptomatic Malaria among Health Care Workers According to Respondents’ Characteristics

This shows how the knowledge of respondents about the existence, definition, diagnosis, and transmission of asymptomatic malaria was distributed according to their ages, gender, professional category, longevity of service, and practicing site.

#### 3.3.1. Participants’ Knowledge about Asymptomatic Malaria by Age

Figure 2a shows that the proportion of respondents who knew how asymptomatic malaria was defined and diagnosed was higher among HCWs younger than 30 years old (<30) than in older respondents, while the proportion of respondents who responded correctly about transmission was higher among HCWs at least 30 years old (≥30) than in younger participants.

#### 3.3.2. Participants’ Knowledge about Asymptomatic Malaria by Gender

Figure 2b shows that the proportion of respondents who were knowledgeable about the existence and transmission of asymptomatic malaria was higher among female than male respondents, while the proportion of correct respondents for the definition and diagnosis of asymptomatic was higher among male than female respondents.

#### 3.3.3. Participants’ Knowledge about Asymptomatic Malaria by Professional Category

Figure 2c indicates that the professional category of nursing assistants had the lowest proportion of asymptomatic malaria-informed respondents.

#### 3.3.4. Participants’ Knowledge about Asymptomatic Malaria by Longevity of Service

Figure 2d shows that the proportion of respondents with knowledge of the definition and diagnosis was higher among HCWs with 0–5 years of practice than among those with more years of practice, while the proportion of respondents with knowledge of existence and transmission of asymptomatic malaria was higher among HCWs with >10 years of practice than among those with fewer years of practice.

#### 3.3.5. Participants’ Knowledge about Asymptomatic Malaria According to Practicing Site

Figure 2e reveals that participants practicing in rural hospitals had a higher proportion of respondents who were knowledgeable about asymptomatic malaria.

### 3.4. Factors Associated with Knowledge of Asymptomatic Malaria among Health Care Workers

Table 3 and Table 4 show participants’ characteristics associated with their knowledge of asymptomatic malaria using logistic regression analysis. Multiple logistic regression analysis showed that the professional category of HCWs was significantly associated with knowledge about the existence of asymptomatic malaria (*p* < 0.05), and longevity of service was associated with knowledge about transmission. Nursing assistants were less likely to be aware about the existence of asymptomatic malaria compared to physicians [OR 0.17, 95%CI 0.03, 0.91] (Table 3); and HCWs with 0–5 years of practice were less likely to be informed about asymptomatic malaria transmission compared to those with more than 10 years of practice [OR 0.24, 95% CI 0.07, 0.85] (Table 4).

### 3.5. Knowledge of Health Care Workers to Use A. afra Tea Infusions to Treat Asymptomatic Plasmodium Carriers

Table 5 reveals that two-thirds 67 (67%) of the respondents knew about *A. afra* tea infusions, they and said it was used to treat malaria (57/67; 85.1%), COVID-19 (34.3%), cough (3%), fever (3%), and others (7.5%) including stomach offset, sinuses, gastritis, typhoid fever, myomas, and diabetes. Most respondents had also heard that *A. afra* was used for the prevention and treatment of acute malaria (58/67; 86.6%) and asymptomatic malaria (42/67; 62.7%).

More than half (38/67; 56.7%) of respondents believed that *A. afra* tea was effective in treating acute malaria, and 20.9% thought that more research was needed, while 9% thought that it was not effective, and 13.4% did not know if it was effective or not. In the same way, 53.7% of respondents believed that *A. afra* tea was an effective treatment for asymptomatic *Plasmodium* carriers, 23.9% thought that more research was needed, 1.5% thought it was not effective, and 20.9% did not know if it was effective or not (Table 5).

### 3.6. Opinions of Health Care Workers about the Use of Artemisia afra Tea to Treat Asymptomatic Plasmodium Carriers

Figure 3 shows that the main opinions and concerns of respondents about the use of *Artemisia afra* tea included a need to ensure the effectiveness of the tea (27/67; 40.3%), a precise posology (26.8), and the absence of side effects (4.5%).

### 3.7. Acceptability of Health Care Workers to Use Artemisia afra Tea Infusion to Treat Asymptomatic Plasmodium Carriers

Figure 4a shows that most of the HCWs 51/67 (76.4%) were willing to prescribe or recommend *Artemisia afra* tea infusions to treat asymptomatic malaria if it was effective (16.2%), it if was approved/recommended by state authorities (13.2%), and it if was available and affordable (10.3%). Others said they would prescribe it because they had tried it before and it worked (10.3%), because they believe it would eliminate resistant forms of malaria (8.8%), because carriers do not have acute malaria (5.9%), because some patients prefer teas (2.9%), and because they believed it was natural with few side effects (2.9%). Meanwhile, 23.2% of respondents said they would not prescribe the teas because they thought that there is no scientific drug data available on the tea (14.7%); it was complicated for patients to take (1.5%); and it could cause resistance in the future (1.5%).

Notwithstanding the aforementioned aspects of acceptability, more than three-quarters (79%) of respondents were ready to prescribe/use *Artemisia afra* teas if endorsed by the state authorities, and 19% said they would not continue to prescribe it after endorsement unless they received knowledge of its prescription, while 2% were still undecided (Figure 4b).

### 3.8. Challenges Pointed Out by HCWs in Using Artemisia afra Tea Infusion for Treatment of Asymptomatic Malaria

Figure 5 shows the challenges of using *Artemisia afra* tea infusions to treat asymptomatic *Plasmodium* carriers predicted by HCWs. These included the problem of dosage and posology (19.4%), the likelihood that patients may resist treatment since they have no symptoms (14.9%), proof of efficacy (11.9%), cost and availability (9%), poor compliance because teas are complicated to take (7.5%), no scientific drug data available (6%), the fact that people do not expect teas in the hospital (3%), and possible counterfeit, since the tea is not well known by the population (1.5%).

## 4. Discussion

Malaria is the most widespread endemic disease in Cameroon [46], and the government of Cameroon has made the fight against malaria a priority, with a highlight in the country’s Health Sector Strategy, as well as the adoption of the High Burden High Impact stratification exercise in the National Malaria Strategic Plan (2019–2023) [47]. There has also been optimistic scale-up of the WHO-recommended control actions towards malaria elimination in Cameroon over the last decades, with focus on vector control, early diagnosis and prompt treatment of symptomatic cases, as well as chemoprophylaxis for high-risk population groups [5]. However, these actions alone are not sufficient to eliminate malaria from many endemic regions as they fail to target asymptomatic *Plasmodium* (gametocyte) carriers who constitute more than 95% of infectious human reservoirs and perpetuate the disease transmission in endemic settings, including Cameroon [11,48]. This cross-sectional survey assessed the knowledge of asymptomatic malaria among HCWs in some rural and urban health facilities in and around Yaoundé, in the Centre Region of Cameroon, and the acceptability of using *Artemisia afra* tea infusions to treat asymptomatic *Plasmodium* carriers. The findings provided herein are reflections of 100 targeted respondents from four hospitals in the Mfoundi and Lekie divisions in the Centre Region of Cameroon.

### 4.1. Knowledge of Asymptomatic Malaria among Health Care Workers

Overall, the results of this study show a high level of knowledge of asymptomatic malaria among HCWs but with some significant gaps, where more than one-tenth (12%) of respondents were not aware of the existence of asymptomatic malaria, 17% did not know the definition of asymptomatic, 25% did not know how asymptomatic malaria was diagnosed, 48.9% were not informed on the transmission of asymptomatic malaria, and 30% did not have knowledge of the right treatment of asymptomatic malaria. This knowledge gap among health care workers in this study is disturbing, as it is a malaria-endemic setting where a majority of *Plasmodium* carriers are asymptomatic [48], with about one-third of medical consultations being suspected malaria cases, and one-fifth of health facility visits comprising laboratory-confirmed malaria cases [3] The knowledge gap observed might be because more than one-quarter (26%) of the survey respondents were of a very low professional category (nursing assistants) and might never have received formal training to consult and treat patients. This is confirmed by other findings of this study, namely, that the Nursing assistants had the lowest proportion of asymptomatic malaria-informed respondents and were less likely to be aware about the existence of asymptomatic malaria compared to higher professional categories with adequate training (*p* < 0.05). As unlicensed professionals, Nursing assistants work with the registered nurses to assist patients with daily care such as ambulating, bathing, feeding, and other typical responsibilities delegated by the registered nurses such as collecting vital signs [49]. Nursing assistants are not normally supposed to perform direct treatment procedures on patients such as invasive procedures, administration of any medications, or the application or removal of any dressings [50,51]. In the current study, the activities of the Nursing assistants went beyond nursing practice to their involvement in patient consultation and treatment prescription, which may have an implication on the quality of patients’ care and safety. In line with this finding, a systematic review on activities of nursing assistants revealed that many activities were performed beyond the training of the nursing assistant and were being performed with limited supervision from registered nurses [52]. Another review in Canada showed that unregulated care providers did not have a scope of practice, but their role had evolved to include activities previously performed by regulated professionals [53].

The involvement of low-category HCWs in patients’ treatment in Cameroon may be due to a high paucity of health care practitioners in Cameroon, with a doctor/patient ratio of 1:50,000, and because about 35,000 health care practitioners are needed to meet the WHO’s recommended ratio of 1 doctor per 10,000 persons [54]. It is therefore impossible for medical consultation and treatment to be performed solely by physicians, hence the inevitable involvement of under-qualified personnel in patients’ consultation and treatment. Given that nursing assistants are often included as a part of the care delivery team and also provide patient care by themselves, it is critical for doctors and registered nurses to share patients’ care information and increase collaboration with them, in order to improve on the quality of patient care provided in such settings. Seminars on asymptomatic malaria should be included in routine in-service training planned for HCWs, with an emphasis on the need to screen and adequately treat all asymptomatic carriers during routine consultations as necessary.

The results that one-quarter (25%) of respondents in this study did not know how asymptomatic malaria is diagnosed and that almost one-third (30%] prescribed conventional antimalarial drugs with non-gametocidal activity to asymptomatic gametocyte carriers is a concern. This is because gametocytes are insensitive to most conventional antimalarial drugs, and they often persist in the blood stream over several months after a successful antimalarial therapy, thereby ensuring continued malaria transmission after the clearance of asexual parasites [12,13]. This results in the widespread occurrence of asymptomatic *Plasmodium* (gametocyte) carriers, who often do not seek antimalarial treatment due to the absence of clinical illness. Thus, there is a need for health care workers to consciously engage in the adequate diagnosis and treatment of asymptomatic carriers with gametocidal drugs, to curb transmission. The knowledge gaps on asymptomatic malaria diagnosis and treatment observed among participants in this study may be because current malaria control measures implemented by the National Malaria Control Program (NMCP) do not target the elimination of asymptomatic *Plasmodium* carriers [5]. The measures in place include use of impregnated mosquito bed nets and indoor insecticide sprays, prophylactic treatment of vulnerable persons, and prompt diagnosis and treatment of symptomatic persons with ACTS and other conventional malaria therapies. It is therefore necessary for the NMCP to include the accurate diagnosis and treatment of asymptomatic *Plasmodium* carriers as a significant aspect of the control and elimination strategic plan, as they substantially contribute to the malaria burden in Cameroon. Due to the emerging resistance to ACTs reported in several African countries [8,9,10], studies have suggested the combination of a gametocytocidal drug such as primaquine with an ACT regimen in clearing gametocytes, to effectively block transmission [55]. In the same light, the WHO recommends that a single dose of 0.25 mg/kg primaquine in combination with ACT administered to patients with *P. falciparum* malaria is safe and effective in reducing transmission without requiring to test for G6PD deficiency [56]. Generally, several studies have been conducted on asymptomatic malaria in Cameroon [48,57,58,59,60,61,62,63] but there are no special programs to target the mass treatment of asymptomatic *Plasmodium* reservoirs through which awareness on the severity of asymptomatic malaria and need for treatment could be raised among health care providers and in the population. Studies reveal that asymptomatic reservoirs can be eliminated in mass treatment programs [64,65], and various mass treatment strategies such as mass drug administration (MDA) have been proposed and proven to rapidly reduce malaria burdens, particularly in regions with seasonal malaria transmission, such as those in Africa and in China [64,65,66].

The finding that duration of practice was significantly associated with respondents’ knowledge about asymptomatic malaria transmission, and HCWs with 0–5 years of practice were less likely to be informed about asymptomatic malaria transmission compared to those with more years of practice, is expected, as HCWs may naturally gain more awareness, experience, and learn more about the condition with an increased number of years of practice. However, these findings were different from the results in Ebonyi State, Nigeria, where the number of years of practice was not significantly associated with doctors’ knowledge of the national guidelines of malaria diagnosis and treatment [67]. The results showed that the proportion of correct respondents in rural hospitals was higher than that in the urban hospitals in all the assessment areas, which might be because most of the respondents in the rural hospitals were physicians (46%) (with only 18% being nursing assistants), while most of the respondents in the urban hospital were nursing assistants (34%).

### 4.2. Opinions, Acceptability, and Challenges of HCWs on Artemisia afra Tea Use

The level of pre-existing knowledge of *Artemisia afra* tea among the survey participants was impressive, as two-thirds of respondents had heard about *Artemisia afra* tea before, and more than four-fifths of them stated that it was used for malaria treatment, while three-fifths had heard about its use in asymptomatic malaria treatment. Hence, introducing the tea will not be completely strange to health care workers, especially with their awareness of its use in malaria treatment. However, the HCWs must be given clarification that the tea is usually recommended only for the elimination of asymptomatic *Plasmodium* reservoirs and not as an alternative acute malaria treatment. The results showed that more than half of respondents believed in the usefulness of the tea in treating symptomatic and asymptomatic malaria, and that more than two-fifths thought that more research was needed to confirm the efficacy of the tea, which is positive feedback, indicating a high potential for acceptability if efficacy is proven. This also shows a high level of pre-existing exposure of the respondents to *Artemisia afra* tea, which could make the tea integration process in the health care system easier.

The results showed that the major concerns of respondents on use of the tea were the need to ensure effectiveness, precise posology, and absence of side effects, which suggests that these aspects should verified when considering its endorsement and integration into the health care system. Even though some of these concerns have been addressed by previous studies that reported the effectiveness of the tea against gametocytes with no toxicity [17,18], there is a need for clinical trials to be conducted to determine the exact efficacy, optimal dosage, and safety of the tea in different population groups, before its official recommendation for use.

The finding that more than three-quarters of respondents were willing to prescribe the tea if its effectiveness, approval by authorities, availability, and affordability was ascertained is a positive opinion with regard to the tea acceptability. Given that the major concern of those who were unwilling to prescribe the tea was logically based on their uncertainty about the availability of scientific drug data on the tea, it is necessary for health authorities to verify and ensure that detailed pharmacology information on the tea is available to health care providers before its endorsement for use in the health care system. The high level of willingness of HCWs to accept the use of *Artemisia afra* tea if endorsed by authorities indicates a promising acceptance and utilization by the population who rely on the decisions of HCWs for their health care. A study on the community perspective of asymptomatic malaria treatment in The Gambia showed high trust of respondents in voluntary health workers and health facility personnel [68].

The predicted challenge of probable treatment refusal by patients due to the absence of disease symptoms is feasible as it is not easy to convince someone to take treatment when they do not feel sick. Previous studies on asymptomatic conditions have shown that uncertain outcomes relating to one not knowing whether they had a disease that was asymptomatic were reasons for treatment refusal or abandonment [69,70]. However, evidence of a positive screening test and a clear justification of the need for treatment by the health care providers may be enough to convince asymptomatic *Plasmodium* carriers for treatment compliance. In an assessment of community perspectives on treating asymptomatic infections for malaria elimination in The Gambia, participants questioned how they could be aware of having the disease when they did not experience any symptoms, given that they had a clear understanding of malaria symptoms, and the lack of these symptoms in the asymptomatic phases led to a state of uncertainty. However, the study population showed a willingness to accept RDT-positive results in the absence of symptoms, and most of them agreed that anti-malarial medications could cure malaria even if the disease was in a dormant phase [68]. Evidence of positive opinions and promising compliance of asymptomatic persons to treatment based on positive diagnostic results highlight the need for authorities to ensure the availability of accurate diagnostic technics for asymptomatic malaria diagnosis and strengthen the capacity of biomedical specialists in the use of these technics. The anticipated challenge of probable poor compliance of potential users because the tea is complicated to take suggests a need to adapt the tea in a form that will be easy to utilize, in order to eliminate poor adherence due to this factor. The risk of counterfeiting can be very common and real as the tea is cultivated locally and can be made available to community members by some farmers who may deliberately and fraudulently add other inactive local herbs or falsify the tea for financial gains. A critical review of the situation of poor quality medicines in Cameroon showed an average prevalence of substandard and falsified (SF) medicines of 26.9%, with about two-thirds of samples from the informal sector, one-fifth from the formal sector, and one-tenth from both sectors, with most of the SF medicines belonging to the anti-infective class [71]. This is dangerous as it may decrease the treatment efficacy, thus resulting in an increased disease burden and missed goal of the treatment. Fake medications may also reduce credibility due to poor treatment outcomes or other adverse effects including disability or death. It is therefore necessary for health authorities to raise awareness and educate the population on the risk of existing counterfeit medications and how to avoid purchasing them. The population should also be encouraged to buy medicines and medical products only from the hospital pharmacy, licensed retail shops, pharmacies, and dispensaries and avoid buying from online pharmacies. They should also be educated on how to visually inspect the medicine’s external packaging to verify that it is labelled with the product name, manufacturer, its expiry date, and the batch number where possible.

To the best of our knowledge, this study is the first to assess the knowledge of asymptomatic malaria among health care workers in Cameroon. However, the study was focused on district health facilities and limited to four health facilities in the Centre Region of Cameroon, with a predetermined sample size, thus limiting the scope of applicability of the results. The addition of primary and tertiary or central health facilities would have been desirable, to assess the variability in care provider knowledge of asymptomatic malaria across the different levels of healthcare. This is therefore recommended in future studies.

## 5. Conclusions

In conclusion, this study revealed a fair knowledge of asymptomatic malaria among health care workers in the survey settings in the Centre Region of Cameroon, with some lapses in their knowledge of diagnosis, treatment, and transmission. Most participants had a positive opinion and willingness to use *Artemisia afra* tea in the treatment of asymptomatic *Plasmodium* carriers if authorized. However, there is a need for health authorities to: organize training seminars on asymptomatic malaria for HCWs; include the accurate diagnosis and treatment of asymptomatic *Plasmodium* carriers as a significant aspect of the malaria control and elimination strategic plan; ensure the availability of accurate diagnostic technics for asymptomatic malaria diagnosis and strengthen the capacity of biomedical specialists in the use of these technics; verify and ensure that detailed pharmacology information on the tea is available to health care providers before its approval for use in health facilities; and raise awareness of the population on the risk of existing counterfeit medications and how to avoid purchasing them.

## Figures and Tables

**Figure 1 ijerph-20-06309-f001:**
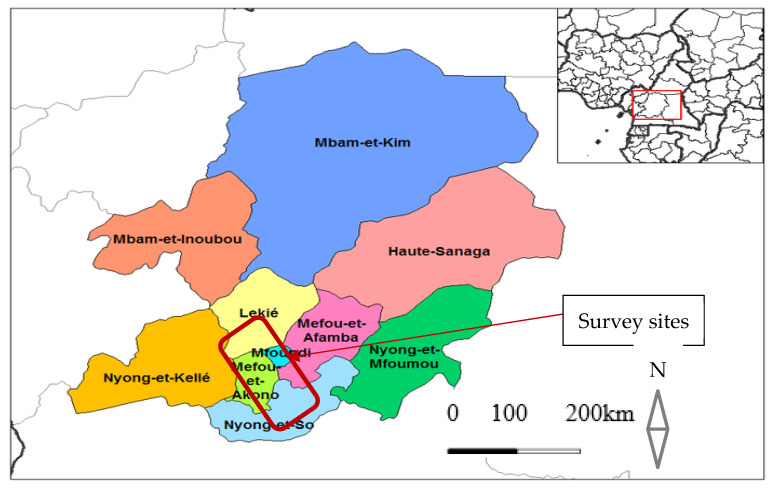
Map of the Center Region of Cameroon showing 10 administrative divisions and the study sites. 4.6298° N; 11.7068° E (Source: Rarelibra 19:57, 2006).

**Figure 2 ijerph-20-06309-f002:**
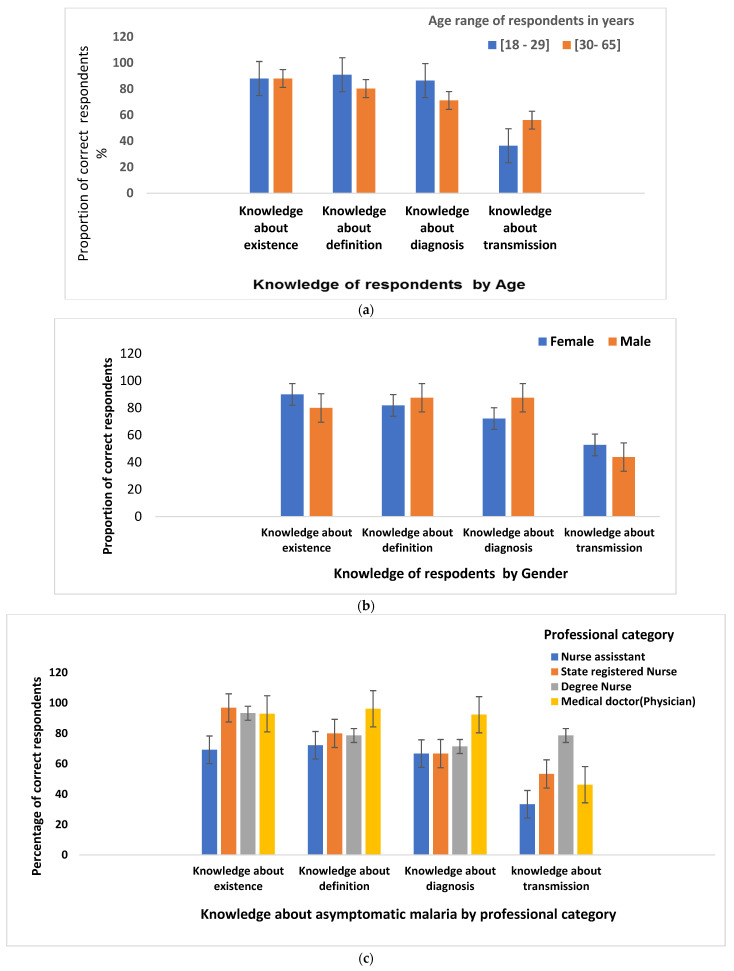
(**a**) Participants’ knowledge about asymptomatic malaria by age; (**b**) participants’ knowledge about asymptomatic malaria by gender; (**c**) participants’ knowledge about asymptomatic malaria by professional category; (**d**) participants’ knowledge about asymptomatic malaria by longevity of service; (**e**) participants’ knowledge about asymptomatic malaria according to practicing site.

**Figure 3 ijerph-20-06309-f003:**
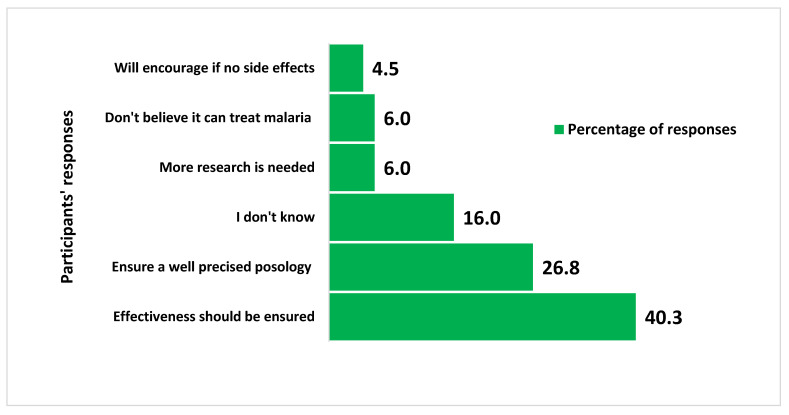
Participants’ opinions on the use *of Artemisia afra* tea to treat asymptomatic *Plasmodium* carriers.

**Figure 4 ijerph-20-06309-f004:**
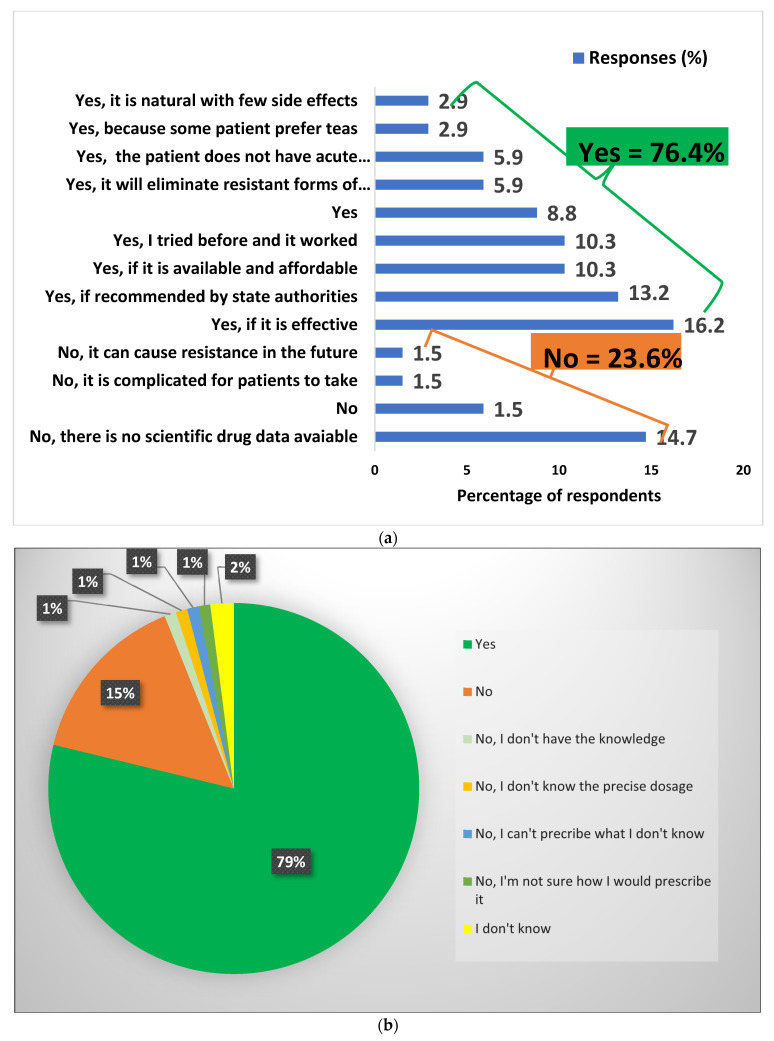
(**a**) Acceptability of HCWs to use *Artemisia afra* tea to treat *Plasmodium* carriers; (**b**) Acceptability of HCWs to prescribe *Artemisia afra* tea if recommended by authorities.

**Figure 5 ijerph-20-06309-f005:**
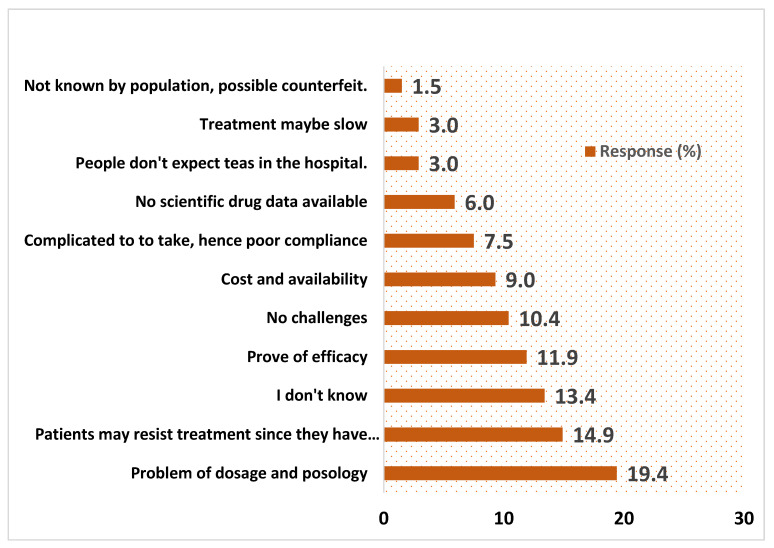
Challenges pointed out by HCWs in using *Artemisia afra* tea infusion for treatment of asymptomatic malaria.

**Table 1 ijerph-20-06309-t001:** Characteristics of survey participants in rural and urban hospitals.

Sample Characteristic	Whole Sample *n* = 100	Rural Hospital *n* = 50	Urban Hospital *n* = 50
	*n* (%)	%	%
Age of respondent (years)			
18–29	25 (25.0)	24.0	26.0
30–65	75 (75.0)	76.0	74.0
Gender			
Female	80 (80.0)	82.0	78.0
Male	20 (20.0)	18.0	22.0
Professional category			
Nursing assistant (NA)	26 (26.0)	18.0	34.0
State registered nurse (SRN)	31 (31.0)	20.0	42.0
Degree nurse (Senior nurse)	15 (15.0)	16.0	14.0
Medical doctor (Physician)	28 (28.0)	46.0	10.0
Longevity of service (years)			
0–5	49 (49.0)	48.0	50.0
6–10	23 (23.0)	16.0	30.0
>10	28 (28.0)	36.0	20.0

**Table 2 ijerph-20-06309-t002:** Knowledge of asymptomatic malaria among health care workers.

Questions Asked	Responses	
	*n*	%
1. Have you ever heard about asymptomatic malaria? (If No, go to section B)		
Yes	88	88.0
No	12	12.0
2. If Yes, what is asymptomatic malaria? (*n* = 88 out of 100)		
Defined asymptomatic malaria correctly	73	83.0
Defined asymptomati malaria incorrectly	15	17.0
3. How can it be diagnosed?		
Knows how to diagnose asymptomatic malaria	66	75.0
Does not know how to diasnose asymptomatic malaria	22	25.0
4. Have you ever seen a patient with asymptomatic malaria?		
Yes	75	85.2
No	13	14.8
5. If Yes, did you put them on treatment? (*n* = 75 out of 100)		
Yes	70	93.3
No	5	6.7
6. If Yes, which drug did you prescribe? (*n* = 70 out of 75) (several options)		
ACT	49	70.0
Artesunate	33	47.1
Quinine	12	17.1
Others	31	44.2
7. Is it necessary to treat asymptomatic malaria?		
Yes, Risk of developing complications	36	40.9
Yes, Risk of becoming symptomic	23	26.1
Yes, We should treat all forms of malaria	10	11.4
Yes, Risk of transmission	2	2.3
No risk if not treated	9	10.2
I don’t know	8	9.1
8. Do you think there is any risk of not treating asymptomatic plasmodium carriers?		
Yes, Risk of developing complications	52	59.1
Yes, Risk of becoming symptomic	16	18.2
Yes, We should treat all forms of malaria	5	5.7
Yes, Risk of transmission	3	3.4
No risk if not treated	4	4.5
I don’t know	8	9.1
9. Can malaria infection be transmitted from a Plasmodium parasite carrier to an uninfected person?		
Yes, because the parasite is present	45	51.1
No, because malaria is not a transmissible disease	31	35.2
No, because it’s asymptomatic and the quantity is mild	7	8.0
I don’t know	5	5.7
10. Can a person infected with parasites from an asymptomatic carrier develop malaria symptoms?		
Yes, because the parasite is present	29	52.7
Yes, because immune systems are different	20	36.4
I don’t know	6	10.9
Missing	33	37.5

**Table 3 ijerph-20-06309-t003:** Factors associated with knowledge about the existence and definition of asymptomatic malaria among HCWs.

Sample Characteristic	Knowledge of Existence of ASM				Knowledge of Definition of ASM					
	*n*	%	OR (95% CI)	*p*-Value	AOR (95% CI)	*p*-Value	*n*	%	OR (95% CI)	*p*-Value
**Total**	100	88.0					88.0	83.0		
** *Age of respondent (years)* **				>0.99						0.23
18–29	25	88.0	1		NI		22	90.0	2.45 (0.51–11.84)	
30–65	75	88.0	1				66	80.3	1	
** *Gender* **				0.24		0.07				0.58
Female	80	90.0	1		1		72	81.9	1	
Male	20	80.0	0.44 (0.12–1.66)		0.16 (0.22–1.15)		16	87.5	1.54 (0.31–7.63)	
** *Professional category* **				**0.01 ***		**0.03 ***				0.11
Nursing assistant	26	69.2	0.17 (0.03–0.91)		0.07 (0.01–0.77)		18	72.2	0.10 (0.01–0.99)	
State registered nurse	31	96.8	2.37 (0.20–26.94)		1.66 (0.68–19.89)		30	80.0	0.16 (0.02–1.43)	
Degree nurse	15	93.3	1.08 (0.09–12.95)		0.57 (0.03–9.32)		14	78.6	0.15 (0.14–1.57)	
Medical doctor (Physician)	28	92.9	1		1		26	96.2	1	
** *Longevity of service (years)* **				0.27						0.36
0–5	49	89.8	0.68 (0.12–3.74)		NI		44	88.6	2.34 (0.64–8.62)	
6–10	23	78.6	0.41 (0.05–1.59)				18	77.8	1.05 (0.25–4.42)	
>10	28	92.1	1				26	76.9	1	
** *Practicing site* **				0.21		0.73				0.10
Rural Hospital	50	78.3	2.19 (0.61–7.80)		1.31 (0.29–5.91)		46	89.1	2.56 (0.80–8.25)	
Urban Hospital	50	71.4	1		1		42	76.2	1	

* Statistically significant *p*-value.

**Table 4 ijerph-20-06309-t004:** Factors associated with knowledge about the diagnosis and transmission of asymptomatic malaria among HCWs.

Sample Characteristic	Knowledge about Diagnosis of ASM				Knowledge of Tranasmission of ASM					
	*n*	%	OR (95% CI)	*p*-Value	*n*	%	OR (95% CI)	*p*-Value	AOR (95% CI)	*p*-Value
Total	88	75.0			88	51.1	83.0			
Age of respondent (years)				0.14				0.11		
18–29	22	86.4	2.56 (0.68–9.67)		22	36.4	0.45 (0.17–1.21)		0.70 (0.21–2.39)	0.57
30–65	66	71.2	1		66	56.1	1		1	
Gender				0.18				0.51		
Female	72	72.2	1		72	52.8	1		NI	
Male	16	87.5	2.69 (0.56–12.93)		16	43.8	0.70 (0.07–0.61)			
Professional category				0.07				0.07		0.12
Nursing assistant	18	66.7	0.17 (0.03–0.95		18	33.3	0.58 (0.17–2.03		0.37 (0.09–1.47)	
State registered nurse	30	66.7	0.17 (0.03–0.85)		30	53.3	1.33 (0.47–3.82)		0.96 (0.30–3.08)	
Degree nurse	14	71.4	0.21 (0.03–1.33)		14	78.6	4.28 (0.96–19.01)		2.98 (0.57–15.74)	
Medicadoctor (Physician)	26	92.3	1		26	46.2	1		1	
Longevity of service (years)				0.11				0.01 *		0.08
0–5	44	84.1	3.30 (1.07–10.23)		44	36.4	0.21 (0.07–0.61)		0.24 (0.07–0.85)	
6–10	18	72.2	1.63 (0.44–5.96)		18	55.6	0.46 (0.13–1.64)		0.37 (0.09–1.48)	
>10	26	61.5	1		26	73.1	1		1	
Practicing site				0.46				0.29		
Rural Hospital	46	78.3	1.44 (0.55–3.79)		46	78.6	1.57 (0.68–3.65)		NI	
Urban Hospital	42	71.4	1		42	46.2	1			

* Statistically significant *p*-value.

**Table 5 ijerph-20-06309-t005:** Knowledge and impression of HCWs on *A. afra* tea.

Questions Asked	Responses	
	*n*	%
1. Do you know about Artemisia afra tea infusions? (If no, go to question 10)		
Yes	67	67.0
No	33	33.0
2. What is it used for? (*n* = 67 out of 100)		
Malaria	57	85.1
COVID-19	23	34.3
Cough	2	3.0
Fever	2	3.0
Others	5	7.5
3. Have you ever heard about its use in malaria prevention or treatment?		
Yes	58	86.6
No	9	13.4
4. Have you ever heard about its use in the prevention or treatment of asymptomatic malaria?		
Yes	42	62.7
No	25	37.3
5. What is your impression about the use of Artemisia afra tea in preventing or treating acute malaria patients?		
It is effective	38	56.7
More research is needed	14	20.9
I don’t know	9	13.4
It is not effective	6	9.0
6. What is your impression about the use Artemisia afra tea infusions in treating asymptomatic plasmodium carriers?		
It is effective	36	53.7
More research is needed	16	23.9
I don’t know	14	20.9
It is not effective	1	1.5

## Data Availability

The data presented in this study are available on request from the corresponding author. The data are not publicly available due to ethical reasons.

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
