# Peer review of "Knowledge about Asymptomatic Malaria and Acceptability of Using Artemisia afra Tea among Health Care Workers (HCWs) in Yaoundé, Cameroon: A Cross-Sectional Survey"

_ijerph, 2023, doi:10.3390/ijerph20136309_

Round 1
Reviewer 1 Report
In the attached file.

Minor editing of English language required.
Author Response
Dear sir/madam ,
We would like to thank you for your useful suggestions which have been fully considered in the revision of the manuscript as indicated in the responses attached.
Yours sincerely.

Reviewer 2 Report
The Manuscript [ijerph-2462902] entitled (Knowledge about asymptomatic malaria and acceptability to use Artemisia afra tea among Health care workers (HCWs) in Yaoundé, Cameroon: A cross sectional survey) assessed the knowledge of asymptomatic malaria (ASM) among health care workers (HCWs) in health facilities (HFs) in the Centre region of Cameroon, and acceptability to use Artemisia afra tea to treat APCs.
Major comments
1- Rearrange figure 2a, 2b, 2d: as follow: [Knowledge of respondents by Age] to be definition of X bar, [Age of respondents in years] to be as indication in upper right. All columns should have the bars of SE, delete numbers from the above of columns
2- As previous comment for figure 2c, 2e but for lines
3- Line 292: Do not use Figure 2a as a title but use valid title which indicate the content. The same in lines 310, 323, 342, 356, 419, 429 , 471
4- Figure 3: Re-design it. It should be as columns, and correct 403 to be 40.3
5- Figure 4a, 4b and figure 5: delete the title above the figure
Minor comments
1- In all MS, write Artemisia afra as italic (as in line 32). Check the whole MS to correct it.
2- Line 90-91: Do not abbreviate the scientific names as (….) but abbreviate them when you mentioned for the second time and after.
3- Line 101: [A. annua] to be italic
4- Line 107: write citation as {7,31] and again in lines 120,122,124, 151 and check all the MS
5- Line 126 and 134: abbreviate the scientific name to be [A. afra]
Minor editing of English language required
See, comments for authors
Author Response
Dear sir/madam'
We would like to thank you for your useful suggestions which have been fully considered in the revision of the manuscript as indicated in the responses attached.
Youes sincerely.

Reviewer 3 Report
The study, subject to expertise, assesses the knowledge of asymptomatic malaria among health workers in health establishments in the Center region of Cameroon. This study also investigates the acceptability of using Artemisia afra tea to treat asymptomatic Plasmodium carriers.
The cross-sectional descriptive survey was conducted among 100 health workers.
The authors indicate that "The questionnaire was developed based on the objectives of the study."; can they specify and indicate how the questions put to the panel were selected.
The study is rich in results but which must be better presented.
This article should be fully proofread for correction of all formatting errors.
The discussion part can be improved by a sharper comparison with the existing one.
Few recommendations;
- Many typographical errors,
- harmonize the presentation of references in the text,
- figures and tables must be introduced by sentences.
In references change "World malaria report 2022 [Internet]. [cited 2023 Mar 22]. Available from: https://www.who.int/publications-detail-723 redirect/9789240064898" by "World malaria report 2022. https://www.who.int/publications-detail-723 redirect/9789240064898
Harmonize the presentation of references.
Author Response

(The authors gave the same response as above.)

Reviewer 4 Report
The manuscript “ijerph-2462902” entitled “Knowledge about asymptomatic malaria and acceptability to use Artemisia afra tea among Health care workers (HCWs) in Yaoundé, Cameroon: A cross-sectional survey” by Qin et al. deals with an interesting subject regarding the assessment of the knowledge of asymptomatic malaria (ASM) among health care workers (HCWs) in health facilities (HFs) in the Centre region of Cameroon, and acceptability to use Artemisia afra tea to treat APCs. A cross-sectional descriptive survey was conducted among 100 HCWs, in four randomly selected HFs in the Centre region between 04/01-20/04/2022, using semi-structured self-administered questionnaires. The results of this study revealed that more than seven in eight (88%) respondents were aware of the existence of ASM, 83% defined ASM correctly, 75% knew how it was diagnosed, 70% prescribed ACTs for APCs and 51.1% were informed about ASM transmission. The professional category of HCWs was significantly associated with knowledge of existence and transmission of ASM and longevity of service was associated with knowledge of transmission (p<0.05). Two-thirds (67%) of respondents knew about Artemisia afra tea, 53.7% believed that it was effective in treating ASM, and 79% were willing to prescribe it if authorized. There was a fair level of knowledge of ASM among HCWs in the study settings.
For publication in the “IJERPH” journal, the topic and content are appropriate. The subject of the manuscript is interesting and topical, with high scientific and practical importance. The introduction is in accordance with the subject and correctly presented. Numerous scientific articles of recent date and in concordance to the topic of the study were consulted. The methodology of the study was clearly presented, and appropriate to the proposed objectives. The obtained results have been fully analyzed and the discussions are appropriate, in the context of the results, and were conducted compared to other studies in the field. The scientific literature, to which the reporting was made, is recent and representative in the field. However, the review of the research article revealed some minor issues, which were noted in the article and listed below:
Minor comments:
· As a scientific name of plant, Artemisia afra must be italicized.
· Keywords: Please change some keywords. The title and keywords must not contain the same words.
· Figures 2a, 2c, and 2d: The text in the figures is liable to be confused. Please fix this problem (e.g., reduce text size).
· References: Authors should correct the form of references as in the journal’s “Instructions for authors”
Thank you for your consideration.
The English language in this manuscript is good.
Author Response
Dear sir/madam'
We would like to thank you for your useful suggestions which have been fully considered in the revision of the manuscript as indicated in the responses attached.
Yours sincerely.

Round 2
Reviewer 2 Report
All my comments were carried out in all sections
Reviewer 3 Report
The authors having improved their article according to my comments and suggestions, I therefore propose the publication.